# Exploring the Potential of Olfactory Receptor Circulating RNA Measurement for Preeclampsia Prediction and Its Linkage to Mild Gestational Hypothyroidism

**DOI:** 10.3390/ijms242316681

**Published:** 2023-11-24

**Authors:** Andréa Harumy de Lima Hirata, Luiz Antônio de Jesus Rocha Camargo, Valdelena Alessandra da Silva, Robson José de Almeida, Lucas dos Santos Bacigalupo, Maria Clara Albejante, Flavia Salomão d’Avila Curi, Patrícia Varela, Leonardo Martins, João Bosco Pesquero, Humberto Delle, Cleber P. Camacho

**Affiliations:** 1Molecular Innovation and Biotechnology Laboratory, Postgraduate Program in Medicine, Universidade Nove de Julho (Uninove), Rua Vergueiro, 235/249, São Paulo 01525-000, SP, Brazil; 2Thyroid Diseases Center, Laboratory of Molecular and Translational Endocrinology, Division of Endocrinology, Department of Medicine, Escola Paulista de Medicina, Universidade Federal de São Paulo, Rua Pedro de Toledo 669, 11th Floor, São Paulo 04039-032, SP, Brazil; 3Department of Obstetrics and Gynecology, Conjunto Hospitalar do Mandaqui, Rua Voluntários da Pátria, 4301, São Paulo 02401-400, SP, Brazil; 4Center for Research and Molecular Diagnostic of Genetic Diseases, Department of Biophysics, Universidade Federal de São Paulo (UNIFESP), Rua Pedro de Toledo 669, 9th Floor, São Paulo 04039-032, SP, Brazil; 5McKusick-Nathans Institute of Genetic Medicine, Johns Hopkins University School of Medicine, Baltimore, MD 21205, USA; 6Division of Medical Sciences, Laboratory of Transcriptional Regulation, Institute of Medical Biology of Polish Academy of Sciences (IMB-PAS), Lodowa 106, 93-232 Łódź, Poland

**Keywords:** mild gestational hypothyroidism, preeclampsia, olfactory receptor, transcriptome, circulating RNA, predictive biomarkers

## Abstract

Gestational hypothyroidism may lead to preeclampsia development. However, this pathophysiological is unknown. We expect to find a shared mechanism by comparing hypothyroidism and preeclampsia. From our transcriptome data, we recognized olfactory receptors as that fingerprint. The reduction of taste and smell in hypothyroid patients has been known for a long time. Therefore, we decided to look to the olfactory receptors and aimed to identify genes capable of predicting preeclampsia (PEC). Methods: An Ion Proton Sequencer (Thermo Fisher Scientific, Waltham, MA, USA) was used to construct the transcriptome databases. RStudio with packages Limma v.3.50.0, GEOquery v.2.62.2, and umap v.0.2.8.8 were used to analyze the differentially expressed genes in GSE149440 from the Gene Expression Omnibus (GEO). The 7500 Real-Time PCR System (Applied Biosystems, Foster City, CA, USA) was used for RT-qPCR amplification of *OR6X1* and *OR4E2*. Results: Our transcriptomic datasets analysis revealed 25.08% and 26.75% downregulated olfactory receptor (ORs) in mild nontreated gestational hypothyroidism (GHT) and PEC, respectively. In the GSE149440 GEO dataset, we found *OR5H1*, *OR5T3*, *OR51A7*, *OR51B6*, *OR10J5*, *OR6C6*, and *OR2AG2* as predictors of early-onset PEC. We also evaluate two chosen biomarkers’ responses to levothyroxine. The RT-qPCR demonstrated a difference in *OR6X1* and *OR4E2* expression between GHT and healthy pregnancy (*p* < 0.05). Those genes presented a negative correlation with TSH (r: −0.51, *p* < 0.05; and r: −0.44, *p* < 0.05), a strong positive correlation with each other (r: 0.89; *p* < 0.01) and the levothyroxine-treated group had no difference from the healthy one. We conclude that ORs could be used as biomarkers at the beginning of gestation, and the downregulated ORs found in GHT may be improved with levothyroxine treatment.

## 1. Introduction

Thyroid dysfunctions are prevalent during pregnancy, affecting approximately 3.47% for subclinical hypothyroidism, 0.50% for overt hypothyroidism, and 2.05% for isolated hypothyroxinemia [1]. Some studies established the connection with preeclampsia, which affect 2–8% worldwide, including a recent meta-analysis, which found an association even with subclinical gestational hypothyroidism [2,3,4,5,6,7]. Even hypoparathyroidism, which has less evidence in the literature, presents preeclampsia as an outcome [6]. However, others do not confirm the risk increment, and this topic remains controversial [7,8,9]. Moreover, doubts persist because preeclampsia’s pathophysiological mechanism remains unknown, making it challenging to establish diagnostic and therapeutic approaches [10,11].

Furthermore, thyroid hormone has a connection with critical points of hypertension pathogenesis: the vascular smooth muscle, renal perfusion, renin-angiotensinogen-aldosterone system, autonomic nervous system, and myocardial contractibility [12]. Likewise, the physiological thyroid stimulus in pregnancy has comparable effects on systemic arterial blood pressure homeostasis. Based on that, its absence, even low, is expected to favor hypertension [13].

The thyroid hormone’s impact on tissues could be measured through gene expression in the blood [14]. Circulating RNA comes mainly from leukocytes and may reflect the thyroid tissue actions in a noninvasive way [15]. Whole blood RNA may infer the thyroid hormone tissue concentrations, but lymphocytes and monocytes mainly express thyroid receptor α (TRα) isoform [15,16]. Therefore, circulating transcripts may be close to TRα predominant tissues, such as the vascular endothelial cells and the heart [17,18]. Nevertheless, TRα is also related to renin–angiotensin system expression, and one or more thyroid-regulated transcripts in the circulation may indicate a thyroid-hypertension relationship [19].

Due to the controversy about the association between gestational hypothyroidism and preeclampsia, we decided to study these two diseases and find similar molecular signatures. We hypothesized that we might unravel a shared mechanism and predict their development. To better characterize that signature, the patients with the two conditions simultaneously were excluded. To our knowledge, no similar study has compared gene expression in gestational hypothyroidism and preeclampsia.

From our transcriptome data, we recognized olfactory receptors as that fingerprint. It attracts our attention because the association between hypothyroidism and the sense of smell has long been known [20]. In 1975, McConnell et al. demonstrated the reduction of taste and smell in hypothyroid patients. They also showed the reversion of the olfactory sensitivity lost by thyroid hormone replacement [21]. Additionally, lately, decreased olfactory sensitivity was noticed even in subclinical hypothyroid patients and improved by levothyroxine treatment [22]. The same anosmic effect was observed in a hypothyroid mouse model and could be prevented using thyroxine [23]. The impact of hypothyroidism on the sense of smell was also well-established in congenital hypothyroidism in newborns, children, and adults [24,25].

The olfactory receptors are the largest gene family in the mammalian genome with 350 genes in humans and approximately 1000 genes in mice [26,27,28]. We aim to compare the differential expressions in gestational hypothyroidism and preeclampsia and evaluate whether the biomarker signature could predict preeclampsia development in an independent population.

## 2. Results

### 2.1. Differentially Expressed Genes in Transcriptome Libraries

The clinical history, physical examination, and laboratory tests from the nine transcriptome library subjects are shown in Table 1.

First, we found 1276 differentially expressed genes (DEG) in mild nontreated gestational hypothyroidism (GHT), among which 1104 were downregulated (86.52%). Second, we found 1185 DEGs in preeclampsia (PEC), among which 1089 genes were downregulated (91.90%). Finally, we compared the PEC versus GHT groups in the last analysis. We found 619 DEGs, among which 320 were downregulated (51.70%) (Figure 1A,B).

### 2.2. Olfactory Receptor Family

In GHT DEG, 396 (31.04%) and, in PEC, 395 (33.33%) were receptor gene types. From these, 320 out of 396 (80.81%) and 317 out of 395 (80.26%) were ectopic olfactory receptors (ORs) gene types, and all of them were downregulated (Appendix A). After, the most abundant receptor types found in GHT were the G protein-coupled receptor (2.78%), taste receptor (1.77%), and trace amine-associated receptor (1.52%). In PEC were G protein-coupled receptor (2.79%), taste receptor (1.78%), killer cell receptor (1.52%), and trace amine-associated receptor (1.52%) (Figure 1C). The ORs represented 25.08% in GHT and 26.75% in PEC from all the DEG.

Comparing PEC versus GHT groups, 162 were receptor coding genes, 125 out of 162 (77.17%) were ORs, followed by killer receptor (3.71%), taste receptor (2.47%), and G protein-coupled receptor (2.47%).

Our previous study found 1656 DEGs in healthy-thyroid pregnancy (HTP) using a healthy nonpregnant woman set, showing the physiologic mechanism regulated by genes in a healthy pregnancy [29]. Three hundred and four ORs found in HTP were similar between GHT and PEC, and all were upregulated in a healthy pregnancy condition (Appendix A).

A total of 195 ORs were shared between GHT and PEC (Figure 1D). Next, to perform the comparison through a hierarchical cluster, we selected the genes in the GHT and PEC list with fold change (logFC) < −3.0, leaving 93 of 195. The logFC from 93 ORs included and the 102 ORs removed are shown in Appendix A. A clustering comparison between the GHT and PEC groups was performed using the 93 selected ORs (Figure 1E).

### 2.3. Levothyroxine Effect Analysis in Olfactory Receptors

We selected two ectopic ORs to measure the *OR6X1* and *OR4E2* based on the cluster analysis; both came from different clusters in the groups (Figure 1E). The clinical history, physical examination, and laboratory tests from 24 subjects (11 HTP, 7 GHT, and 6 levothyroxine-treated patients—GHT-LT4) were selected for the levothyroxine effect analysis evaluation and are shown in Table 2. The patients treated with levothyroxine were administered a dose of 89.58 ± 63.45 mcg/day and Thyroid Stimulating Hormone (TSH) up to 4.0 mIU/L. Unfortunately, we could not detect *OR6X1* expression in four blood samples and the *OR4E2* expression in only one sample.

In addition, we observed different values of *OR6X1* RE (*p* < 0.05) and *OR4E2* RE (*p* < 0.05) between HTP and patients of GHT groups (Figure 2A,B). There was no difference between the HTP and levothyroxine-treated patients (GHT-LT4) groups for both genes. Moreover, we observed a strong correlation between *OR6X1* and *OR4E2* (r: 0.89; *p* < 0.01). There was a negative correlation between TSH and *OR6X1* (r: −0.51; *p* < 0.05) and between TSH and *OR4E2* (r: −0.44; *p* < 0.05). *OR6X1* and *OR4E2* presented an excellent capability to identify patients with a higher TSH with an AUC: 0.88 and 0.76, respectively.

### 2.4. Preeclampsia Onset Prediction Using Olfactory Receptors

We selected 12 early preeclampsia blood samples and 12 healthy pregnancies as the control group before 20 weeks of gestation from GSE149440 (Figure 3). The average and standard deviation of gestational age in early preeclampsia samples collected were 14.6 ± 1.8, and in the control group were 14.8 ± 2.9. In the early preeclampsia and control group, the gestational age at delivery was 31.4 ± 2.4 and 39.3 ± 1.1, respectively. The gestational age of preeclampsia diagnosis was 30.8 ± 2.6.

We found 12 ectopic ORs differentially expressed with adjusted *p* < 0.05. The 12 ORs we used to evaluate the preeclampsia outcome before 20 weeks of gestational age have a high prediction capability, as seen in Table 3. Furthermore, the combination of all ORs was shown to have excellent diagnostic power. As many as 10 out of 12 were shared with our GHT and PEC transcriptome (*OR5H1*, *OR5T3*, *OR51A7*, *OR51B6*, *OR2W1*, *OR5B17*, *OR11G2*, *OR10J5*, *OR6C6*m and *OR2AG2*), and 7 of them appear in hierarchical ectopic ORs clustering (Figure 1E). Furthermore, these seven ORs showed the capability to predict the occurrence of early preeclampsia (Figure 3). Moreover, the low expression found in GSE149440 showed an increased risk of developing PEC (*OR5H1*, *OR5T3*, *OR51A7*, *OR10J5*, *OR6C6*), and the overexpression was protective (*OR51B6*, *OR2AG2*) (Table 4).

## 3. Discussion

Our study found a link between circulating ORs mRNA in mild gestational hypothyroidism and a similar pattern observed in preeclampsia patients. Other researchers have previously linked thyroid dysfunction and OR outside the olfactory epithelium, and a connection may exist between both systems, favoring some outcome. For example, Massolt et al., in a whole blood transcriptome from levothyroxine-treated patients, encountered the *OR2W3* upregulated [15]. Therefore, as expected, in our study, where we saw the deficiency of thyroid hormone, this gene was downregulated. Moreover, Flegel et al. described the very high expression of OR2W3 in the thyroid [30].

The fundamental role of olfactory receptors in normal physiology is still unknown, but its expression has already been demonstrated in different tissues [30]. They are recognized for their role in sensing small metabolites and gases, which has led to their association with hypoxia studies [31]. Although we do not have a complete picture of the pathophysiology of preeclampsia because it is multifactorial, the incomplete invasion of fetal trophoblasts, followed by a defect in the formation of placental vessels leading to placental ischemia, which promotes adaptive measures in response to hypoxia, is well-established [32].

The connection of hypoxia with hypertension by stimulating the sympathetic system through the olfactory receptors has also been described [33]. Although ORs do not affect the blood pressure baseline, they modulate the renin system [34,35]. ORs’ presence in hypothalamic chemosensors responsible for vasopressin secretion may indicate another related hemodynamic mechanism [36]. According to Shepard B.D. and Pluznick J.L., ORs must be used as chemosensors to maintain homeostasis in the kidney. They also showed that animal OR activation (Olfr78) increased the renin release by short-chain fatty acids (SCFAs) [37]. Pluznick et al. demonstrated the presence of some ORs expressed in the kidney [38]. In our analyses, one of them, *OR2H2*, a homolog of Olfr90, appears as a downregulated DEG in the GHT and PEC groups. It may be exerting a compensatory mechanism against the blood pressure increase.

The connection of olfactory receptors with hypoxia, which is known to increase the expression of type 3 deiodinase (DIO3), could partly explain the association of hypothyroidism with preeclampsia because its pathophysiology is similar to Nonthyroid Illness Syndrome (NTIS) [39]. In NTIS, the T4 conversion to reverse T3 reduces T3 concentrations. As a matter of fact, any situation that could lessen the T3 concentrations, such as in the Thr92Ala deiodinase type 2 genetic variant carriers, in which the risk of preeclampsia increases by more than six times [40].

Beyond hypoxia, other OR physiological functions can generate pathophysiological impacts; some are related to mechanisms predisposing to hypertension. For example, they are associated with Ca^2+^ signal transduction pathways, increasing ion concentration [41,42]. Growth factors and cytokines are present at inadequate concentrations during preeclampsia, leading to endothelial dysfunction by inhibiting Ca^2+^ signaling [43]. Zhou et al. showed that serum Ca^2+^ was lower in pregnant women with subclinical hypothyroidism combined with gestational diabetes mellitus than in a healthy pregnancy [44]. In the GHT and the PEC group, all ORs were downregulated, likely leading to endothelial dysfunction and even increased blood pressure due to a reduced mechanism of Ca^2+^ signaling.

The chromosome 14 region where *OR4E2* is located is co-amplified with human epidermal growth factor 2 (*HER2*), which is overexpressed in breast cancer [45]. Meinhardt et al. observed the amplification of HER2 in the differentiation of human extravillous trophoblasts (EVTs) [46]. Triiodothyronine (T3) is shown to influence the EVTs invasion into the decidua, and in breast cancer-positive HER2, free triiodothyronine (FT3) (even in normal concentration) is shown to be negatively correlated with cell proliferation (ki67) [47,48]. The ORs might be related to placentation via HER2 and T3. Consequently, this relationship might influence preeclampsia appearance in hypothyroidism.

Additionally, angiogenesis is fundamental during pregnancy [49]. Kim et al. demonstrated that umbilical vein endothelial cells, the human aorta, and the coronary artery express an OR type, OR10J5, a regulator of angiogenesis [50]. *OR10J5* was among the 93 OR genes shared between the GHT and PEC groups and downregulated in both conditions, which might be reflecting a drop in this angiogenesis, locally or systemically.

In some tissues, such as the kidneys, ORs may also receive insufficient concentrations of exogenous chemicals because it is more internalized [42]. Therefore, it is necessary to identify endogenous agonists, such as hormones [42]. Turunen et al. described that levothyroxine treatment reduces the odds of gestational hypothyroidism of many adverse events, such as hypertension and severe preeclampsia [4]. Additionally, we observed that the RE values of *OR6X1* and *OR4E2* remained similar to the healthy pregnancy group in levothyroxine-treated patients, possibly indicating a benefit of the drug. However, the mechanism between levothyroxine and ORs remains unknown.

Although the association between gestational hypothyroidism and preeclampsia is controversial, we developed a noninvasive tool with which to predict the last with respect to both transcriptomes. The high frequency of downregulated olfactory receptor-like genes between the two diseases caught our attention. Moreover, we found some circulating ORs to predict preeclampsia onset occurrence before 20 gestational weeks. Moreover, pregnant women with loss of expression showed an increased risk of PEC development, and overexpression showed a protective effect.

We understand that our study has some weaknesses. As patient selection was carried out in a tertiary hospital, which only receives highly complex patients, many patients could not be included due to the exclusion criteria. Even though the OR is the largest gene family in mammals, this concern is not entirely appropriate for our study since we used RNA-Seq libraries with limited targets. Additionally, we treated the results with proper analyses using kappa and positive and negative likelihood to correct any error linked to a prevalence deviation associated with the small sample number. We also adjusted the *p*-value for false discoveries, a conservative strategy [51]. We also used the RT-qPCR evaluation to confirm two OR different expression levels between the HTP and GHT groups. Although the sample size is small, and more studies are necessary, the expression was restored via levothyroxine use.

Finally, we emphasize that our transcriptome libraries, the RT-qPCR experiment, and the GSE149440 use circulating RNA data, which favors its use as a biomarker. We believe we controlled for study errors as much as possible and unveiled a connection between the olfactory system and hypothyroidism or preeclampsia.

## 4. Methods

### 4.1. Population

All the participants signed written informed consent. The Institutional Review Board linked to the National Research Ethics Commission (CONEP) from the university outpatient clinic and tertiary public health system hospital approved the study (numbers/CAAE: 665.331/30746814.4.0000.5511, 679.727/30746814.4.3001.5551 and 259.222/15477713.1.0000.5511). The study followed the principles of the Declaration of Helsinki. All the procedures followed were in accordance with institutional guidelines.

### 4.2. Transcriptome Libraries

We used two sets of our transcriptome libraries, HTP and GHT, available at the National Center for Biotechnology Information (NCBI) GEO DataSets database (Accession number: GSE147527). The patients in the GHT group have not initiated levothyroxine treatment.

We constructed a third set of three libraries from patients with PEC (Accession number: GSE157148) in the same period and following the same transcriptomic protocol available below. The PEC transcriptome libraries were constructed from pregnant women with a gestational age greater than 20 weeks. The SBP was ≥140 mmHg, and the DBP was ≥90 mmHg. The PEC diagnosis was made according to the 2013 American College of Obstetricians and Gynecologists task force recommendations (ACOG) [52]. Previously associated pathologies, including diabetes mellitus, chronic arterial hypertension, kidney disease, systemic lupus erythematosus, phospholipid antibody syndrome, licit or illicit drug users, and patients with declining glomerular filtration rate, were excluded.

Pregnant women with TSH ˂ 3.0 mIU/L were assigned to the HTP group. The participants could be pregnant women from any trimester and could present single or twinning live fetuses. They could be primigravida or multigravida and aged older than 18 years. The presence of obstetric conditions that could cause maternal–fetal harm was excluded. The transcriptome participants did not present any medical conditions during pregnancy or were users of alcohol, illicit or licit drugs, which may influence thyroid function. The GHT group should have a TSH level of ˃3.0 mIU/L and less than 10 mIU/L. GHT patients already treated with levothyroxine were included and identified as the GHT-LT4.

### 4.3. Blood Sample Collection and Processing

The PEC samples were collected after 20 weeks of gestation, which allowed the disease confirmation. The blood samples for hormonal analyses for the nine transcriptome libraries were prepared using using the Cobas Roche Elecsys 600 instrument (Roche Diagnostics, Indianapolis, IN, USA) and the Elecsys 2010 Roche Diagnostics instrument (Roche Diagnostics). All the transcriptome participants were negative for antithyroid antibodies at the time of blood sampling.

The TSH concentration was determined using the Abbott Architect I 2000 instrument (Abbott Diagnostics, Abbott Park, IL, USA) in the levothyroxine effect analysis patients. The Abbott TSH has a detection sensitivity of 0.0038 mIU/L and an interassay variance of 3.59%. The free thyroxine (FT4) measurement was performed using Cobas Roche Elecsys 600 instrument (Roche Diagnostics).

### 4.4. RNA Extraction and cDNA Synthesis

We used the PAXgene Blood RNA tube (Qiagen/BD, Venlo, The Netherlands) and the PAXgene Blood RNA Kit (Qiagen, Venlo, The Netherlands). The RNA was stored at −80 °C. The Qubit fluorometer 2.0 (Thermo Fisher Scientific, Waltham, MA, USA) was used in transcriptome and NanoDrop 2000 (Thermo Fisher Scientific, Waltham, MA, USA) in the levothyroxine effect samples analysis.

For cDNA synthesis in the transcriptome, we used SuperScript VILO cDNA Synthesis Kit (Thermo Fisher Scientific, Waltham, MA, USA, catalog number: 11754050), DNase treatment for an RNA concentration of 10 ng, and completed it with nuclease-free water for a total of 5 μL. In RT-qPCR, for the levothyroxine effect samples analysis, we used 4 μL of SuperScript VILO Master Mix (Thermo Fisher Scientific, Waltham, MA, USA, catalog number: 11755050), RNA concentration of 20 ng/μL, and completed it with DEPC water for a total volume of 20 μL. The cDNA was stored at −20 °C until the analysis.

### 4.5. Transcriptome Library Construction

We used the Ion AmpliSeq Library Kit Plus (Thermo Fisher Scientific, Carlsbad, CA, USA) and target regions from the Transcriptome Human Gene Expression panel with 20,802 initial targets, according to the manufacturer’s recommendations. The Ion PI Hi-Q OT2 Reagent 200 (Thermo Fisher Scientific, Frederick, MD, USA) in One Touch 2 Instrument was used for emulsion PCR. We used Ion OneTouch ES (Thermo Fisher Scientific, Carlsbad, CA, USA) with Dynabeads MyOne Streptavidin C1 Beads (Ion Proton™ System, Thermo Fisher Scientific, Vilnius, Lithuania) to enrich template-positive ISPs. The sequencing was performed using an Ion PI Chip (Thermo Fisher Scientific, Taiwan, China) in an Ion Proton Sequencer (Thermo Fisher Scientific, Carlsbad, CA, USA).

### 4.6. Preeclampsia Onset Prediction

We are interested in prediction biomarkers, and for that, we used the GEO database (ncbi.nlm.nih.gov/geo accessed on 10 February 2022) to find genes expressed in blood samples from early preeclampsia before the diagnosis. We selected datasets in which the blood samples were collected before 20 weeks of gestational age. The GSE149440 was the unique databank that matched those parameters. Tarca et al. published this dataset data analysis. They described the study as a prospective longitudinal study in which the blood samples were collected at the beginning and end of pregnancy. They used the 2002 ACOG’s practice guideline and the early preeclampsia was defined when the diagnosis occurred before 34 gestational weeks [53]. RStudio with packages Limma (version 3.50.0), GEOquery (version 2.62.2), and umap (version 0.2.8.8) were used to analyze the differentially expressed genes in GSE149440 from the Gene Expression Omnibus (GEO). The adjusted *p* < 0.05 was used to select significant ORs.

### 4.7. Levothyroxine Effect Analysis

The independent set of healthy pregnant participants and gestational hypothyroidism patients were selected to evaluate the gene expression levels in both situations. Moreover, we included patients with and without levothyroxine treatment to assess the thyroid hormone replacement effect on expression levels. Because of the recent guidelines recommendations, we could only have a small sample of patients treated with levothyroxine in our study conditions [54].

RT-qPCR was used to detect target gene expression levels. We used a 0.5 μL aliquot of cDNA in 10 μL polymerase chain reaction (PCR) amplifications containing 1.25 μL of TaqMan™ Universal Master Mix II (Life Technologies, Thermo Fisher Scientific; catalog number: 4440038) and 0.125 μL of each primer (Life Technologies, Thermo Fisher Scientific; catalog number: Hs02339167_s1 and Hs01102077_s1). The 18S rRNA was used as the reference gene (Life Technologies, Thermo Fisher Scientific; catalog number: Hs99999901_s1).

According to the manufacturer’s protocol, the 7500 Real-Time PCR System (Applied Biosystems, Foster City, CA, USA) was used for RT-qPCR amplification. The reactions were accomplished in 40 cycles, and each sample was performed in triplicate. We used the 7500 Applied Biosystems software version 2.3 (Applied Biosystems) with the averaged SD ≤ 1 to form the threshold cycle. The target mRNA expression was normalized using blood samples collected from healthy-thyroid pregnant patients. The relative expression (RE) of target mRNA was calculated via the formula 2^(Cs − Rs)^/2^(Cn − Rn)^, where Cs is the average of the target mRNA-Cq triplicate of each sample, Rs is the average of the 18S-Cq triplicate of each sample, Cn is the average target mRNA-Cq value normalized by healthy-thyroid pregnancy, and Rn is the average of 18S-Cq triplicates of the same sample.

### 4.8. Bioinformatic and Statistical Analysis

#### 4.8.1. Bioinformatic

We constructed a single-count matrix in a csv type file, unifying each of the nine transcriptome libraries’ reads. The transcriptome matrix was used for analysis using R statistical software v.3.4.1, available at (https://www.R-project.org accessed on 14 February 2020) [55]. The quality control and the variation analysis were conducted using the pcaExplorer package v.20.0 [56].

We performed a three-group analysis among HTP, GHT, and PEC. The trimmed mean of M-values (TMM) from the edgeR package v.3.16.5 was used for library size normalization [57]. We called the differentially expressed genes (DEG) using the same package, and only genes with *p*-values < 0.05, corrected via the False Discovery Rate (FDR) method, were considered significant. We selected the relevant DEGs by the fold change (logFC) cutoffs of +1.0 and −1.0 from the FDR adjustment significant list of transcripts from the groups, showing us the most relevant genes. However, as we were looking for biomarkers, we selected the OR-relevant transcript in GHT and PEC with a logFC cutoff < −3.0.

To construct the Vulcano Plot, we used ggplot2 (version 3.3.5), dplyr (version 1.0.7) and RColorBrewer (version 1.1-2) packages. We constructed the Heatmap using pheatmap (version 1.0.12) and RColorBrewer (version 1.1-2) packages. We used InteractiVenn to detect overlapping genes [58]. A hierarchical clustering comparison between GHT and PEC was performed in R statistical software v.3.4.1. We used the Euclidean distance to perform hierarchical clustering in a three-group analysis and construct the dendrogram. Both were produced using the tanglegram function available in the dendextend package [59].

#### 4.8.2. Statistical Analysis

The data were presented as medians with minimum and maximum values. The sample size was calculated using G*Power v.3.1.9.4 software [60]. The biochemical and molecular data were log-transformed before statistical analysis, except for the anti-TPO antibody with a detection range limitation. Based on the detection limited range, the Kruskal–Wallis test was used for anti-TPO. One-way ANOVA was used to compare the continuous variables in the transcriptome and the levothyroxine effect analysis.

Cox regression was used to calculate the Hazard Ratio. The Kaplan–Meier method was used with the log-rank test for the preeclampsia onset analysis. In GSE149440, we consider the follow-up time as the gestational age of delivery in the control and the gestational age of diagnosis in the preeclampsia group. The sensitivity, specificity, positive predictive value (PPV), negative predictive value (NPV), and accuracy were calculated with the Galen and Gambino formulae [61]. To calculate the reliability, we used Cohen kappa (κ). We also calculated the positive and negative likelihood (PL and NL) that allowed us to observe the test diagnostic capability without the influence of the prevalence of the disease on the datasets. The excellent diagnostic power is reflected by a positive likelihood higher than one and a negative likelihood near zero.

Tukey’s post hoc analysis was used to compare the RE between the groups. A chi-squared test was used to compare the frequencies and Fisher’s exact test was applied when necessary. The correlation was performed via Pearson’s method. We constructed an ROC curve to obtain the Area Under the Curve (AUC) and categorize variables using Youden’s method to establish the cutoff in GSE149440 [62]. The Artificial Intelligence Neural Network (ANN) was constructed to combine the ORs’ relative expression from GSE149440. We set one hidden layer with a hyperbolic tangent function, the batch training type, and randomly assigned 70% of the samples to the training step and 30% to the test. We constructed the ROC curve through the ANN and analyzed the independent variable importance. IBM SPSS Statistics for Windows, Version 28.0, from IBM Corp., released in 2021 (Armonk, NY, USA), was used to analyze the data. A *p*-value < 0.05 was considered significant.

## Figures and Tables

**Figure 1 ijms-24-16681-f001:**
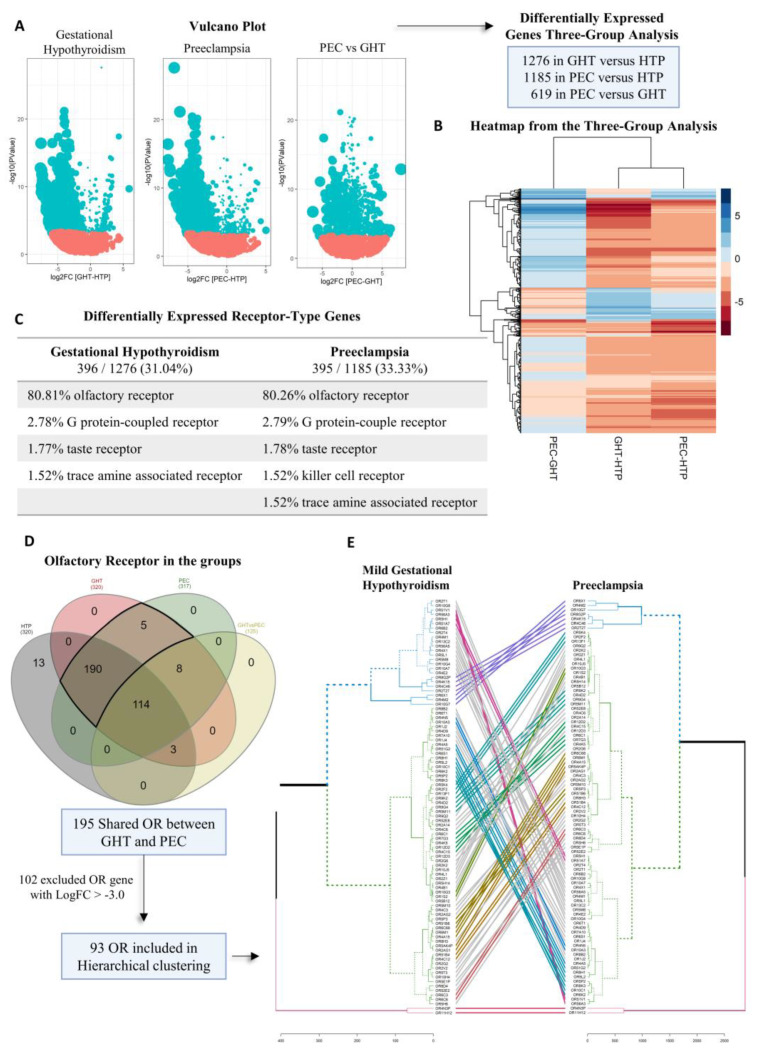
Schematic workflow of differentially expressed genes and ectopic olfactory receptors in blood’s transcriptome. We collected blood samples from healthy-thyroid pregnancy (HTP), mild nontreated gestational hypothyroidism (GHT), and preeclampsia (PEC) to construct the transcriptome and for hormonal concentration measurement. The blood sample collections and transcriptome construction were made in the same period with the same protocol. (**A**) Vulcano Plot presents significant differential genes considering the false discovery rate (FDR) adjustment on the *p*-value in blue dots. The red dots represent the non-significant genes. (**B**) Heatmap demonstrating the comparison among the 1276, 1185, and 619 differentially expressed genes (DEG) from the three-group analysis. The first column shows the difference between preeclampsia versus gestational hypothyroidism (PEC-GHT); in the second, the difference between gestational hypothyroidism versus healthy-thyroid pregnancy, our control (GHT-HTP); and in the last column, the difference between preeclampsia versus healthy-thyroid pregnancy (PEC-HTP). The blue lines are the upregulated and the red are the downregulated genes. (**C**) Table with the percentages of receptor types found in our DEG. (**D**) Venn plot shows the shared ectopic ORs genes among the groups. (**E**) The comparative hierarchical cluster plot, constructed after removing 102 genes with a fold change (logFC) > −3.0 (93 ORs remained); the hierarchical clustering comparison of the ORs shared between the GHT and PEC groups. The coloured lines in the center show where similar ORs are in the cluster.

**Figure 2 ijms-24-16681-f002:**
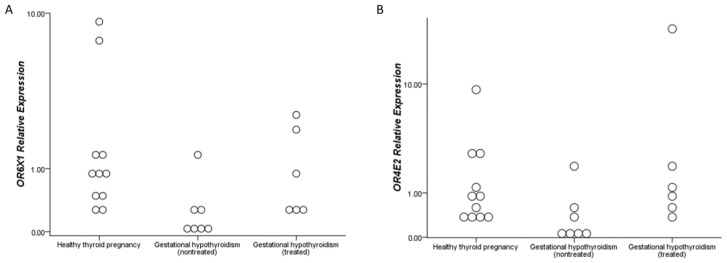
Levothyroxine effect on mild gestational hypothyroidism. 2-D Dot Plot shows *OR6X1* (**A**) and *OR4E2* (**B**) relative expression between healthy-thyroid pregnancy (white) and mild nontreated (dark gray) and treated gestational hypothyroidism (light gray), with the dots representing each participant.

**Figure 3 ijms-24-16681-f003:**
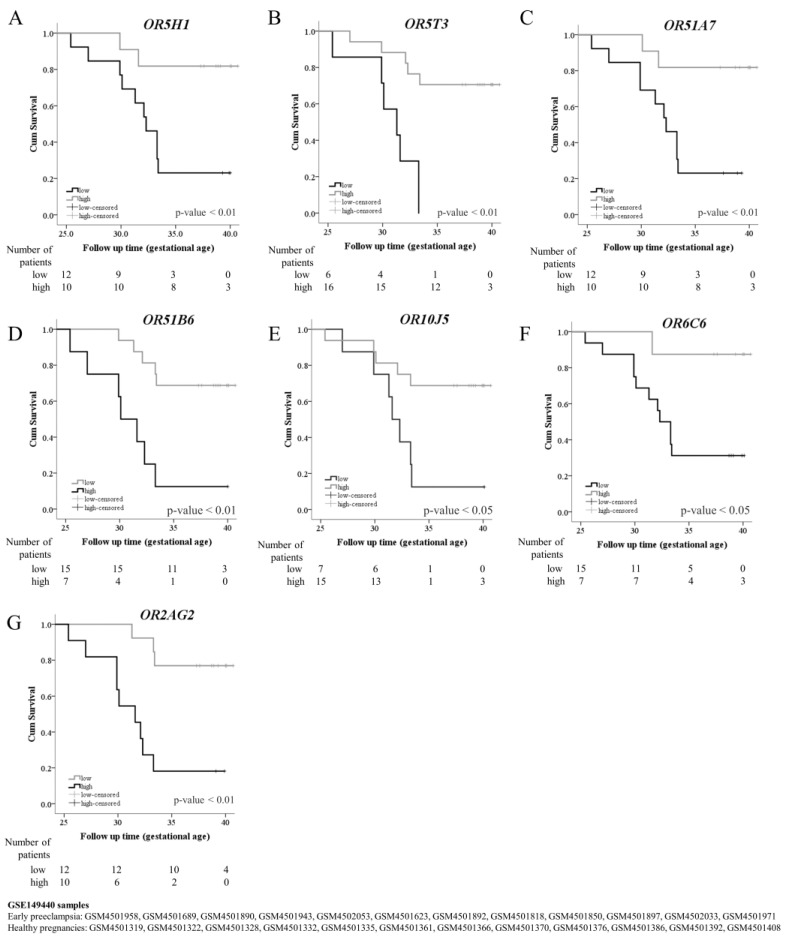
Olfactory receptor predicting preeclampsia onset. Survival curve using gene expression from GEO GSE149440 dataset for the following genes *OR5H1* (**A**), *OR5T3* (**B**), *OR51A7*(**C**), *OR51B6* (**D**), *OR10J5* (**E**), *OR6C6* (**F**), and *OR2AG2* (**G**). We selected the libraries from blood samples collected from patients before 20 weeks of gestation. To set the time, we considered the gestational age of delivery in the control and the gestational age at diagnosis in the preeclampsia group. The preeclampsia diagnosis was used as the outcome, and the relative expression was categorized by the Youden method cutoff as the factor to construct the Kaplan–Meier plot.

**Table 1 ijms-24-16681-t001:** General epidemiological data from transcriptome subjects. Clinical data, physical exams, and laboratory tests from healthy-thyroid pregnancy (HTP), mild non-treated gestational hypothyroidism (GHT), and preeclampsia (PEC). The data are presented in median, maximum and minimum values, or frequency. The *p*-value, except for categorical variables (Chi-square or Fisher’s exact test) and AntiTPO (Kruskal–Wallis test), were log-transformed before ANOVA.

	Healthy-Thyroid Pregnancy	Non-Treated Mild Gestational Hypothyroidism	Preeclampsia	*p*-Value
**Clinical history**				
Age (years)	23 (22–26)	35 (22–38)	31 (29–33)	0.192
Gestational age (weeks)	38.7 (38.4–40.7)	41 (37.2–41.0)	34.5 (32.7–39.8)	0.164
Previous abortions	yes (66.7%)	no (100%)	yes (66.7%)	0.165
Twinning	no (100%)	yes (33.3%)	no (100%)	0.325
Cesarean birth	1 (33.3%)	2 (66.7%)	3 (100%)	0.223
Vaginal delivery	2 (66.7%)	1 (33.3%)	-	0.223
Primigravida	1 (33.3%)	2 (66.7%)	-	0.223
Multigravida	2 (66.7%)	1 (33.3%)	3 (100%)	0.223
Newborn weight (g)	3515 (2695–3850)	3290 (2595–3600)	2730 (2575–3190)	0.488
**Physical exam**				
Weight (Kg)	70.0 (66.0–70.6)	87.5 (63.0–92.0)	86.8 (67.0–102.0)	0.396
SBP (mmHg) *	110 (110–120)	118 (90–120)	140	0.038
DBP (mmHg) *	60 (60–80)	68 (60–80)	110 (100–110)	0.008
**Laboratory tests**				
TSH (µIU/mL) *	1.94 (1.69–2.86)	4.36 (3.70–4.49)	3.12 (1.74–3.29)	0.049
FT4 (µIU/mL)	1 (0.93–1.00)	1 (0.82–1.00)	0.84 (0.77–1.14)	0.797
AntiTPO (UI/mL)	<5	5 (<5–7.27)	<5	0.368

* *p* < 0.05.

**Table 2 ijms-24-16681-t002:** General epidemiological and experimental data. Clinical history, physical exam, laboratory test, and circulating RNA measurement in levothyroxine effect analysis from OR6X1 and OR4E2. The validation group were healthy-thyroid pregnancy (HTP), mild nontreated gestational hypothyroidism (GHT), and treated GHT (GHT-LT4). The data are presented in median, maximum, and minimum values or frequency. The *p*-value for categorical variables was calculated via Chi-square or Fisher’s exact test; the others were log-transformed before ANOVA.

	HTP	GHT	GHT-LT4	*p*-Value
**Clinical history**				
Age	25 (18–39)	23 (19–35)	30.5 (18–43)	0.341
Gestational age ^a^	39.4 (37.1–43.0)	39.8 (30.0–43.0)	32.3 (6.8–41.0)	0.030
Previous abortions	yes (27.3%)	yes (14.3%)	yes (33.3%)	0.711
Twinning	no (100%)	yes (14.3%)	no (100%)	0.282
Primigravida	3 (27.3%)	3 (42.9%)	3 (50%)	0.606
Multigravida	8 (72.8%)	4 (57.2%)	3 (50%)	0.614
Alcoholist	no (100%)	no (100%)	no (100%)	-
Smoker ^a^	no (100%)	no (100%)	yes (33.3%)	0.038
**Physical exam**				
Weight (Kg)	71.0 (52.0–90.0)	83.9 (63.0–143.5)	77.1 (50.9–102.6)	0.251
SBP (mmHg)	110 (90–144)	110 (100–130)	117 (100–130)	0.922
DBP (mmHg)	72 (54–90)	70 (60–89)	73 (60–80)	0.968
Laboratory tests				
TSH (µIU/mL)	2 (0.56–2.94)	4.16 (3.35–7.38)	1.83 (1.08–4.02)	0.001
FT4 (µIU/mL)	0.82 (0.64–1.12)	0.93 (0.80–1.08)	0.75 (0.64–0.98)	0.243
**Relative Expression data**				
*OR6X1* ^b,c,d^	0.831 (0.267–9.027)	0.062 (0.007– 1.352)	0.637 (0.314–2.605)	0.002
*OR4E2* ^a,b,c^	0.824 (0.323–9.085)	0.116 (0.006– 2.101)	0.957 (0.289–25.107)	0.014

^a^ *p*-value < 0.05; ^b^ *p*-value < 0.01; ^c^ *p*-value < 0.05, between mild nontreated gestational hypothyroidism and healthy-thyroid pregnancy. ^d^ *p*-value < 0.05, between mild nontreated gestational hypothyroidism and treated mild gestational hypothyroidism.

**Table 3 ijms-24-16681-t003:** Clinical predictive parameters of each OR alone or combined from the samples collected in the first trimester. The Area Under the Curve (AUC) from an ROC curve, the OR RE cutoff value, direction, expression, chi-square (X2) test, clinical predictive parameters, Cohen kappa (κ) and positive and negative likelihood (PL and NP) calculated for the binary outcome of developing or not developing preeclampsia after 20 weeks of gestational age.

Parameters	AUC	Cutoff	Expression	chi-Square		Sensitivity	Specificity	PPV	NPV	Accuracy	κ	PL	NL
				X2	*p*-Value								
OR5H1	0.847	2.60	Loss	8.224	0.004	0.83	0.75	0.77	0.82	0.79	0.58	3.33	0.22
OR5T3	0.819	2.28	Loss	9.882	0.005	0.58	0.96	0.94	0.69	0.77	0.54	15.00	0.44
OR51A7	0.819	2.74	Loss	8.224	0.004	0.83	0.75	0.77	0.82	0.79	0.58	3.33	0.22
OR51B6	0.771	3.18	Gain	6.750	0.027	0.58	0.92	0.88	0.69	0.75	0.50	7.00	0.45
OR10J5	0.764	2.83	Loss	6.750	0.027	0.58	0.92	0.88	0.69	0.75	0.50	7.00	0.45
OR6C6	0.764	2.67	Loss	6.750	0.027	0.92	0.58	0.69	0.88	0.75	0.50	2.20	0.14
OR2AG2	0.785	4.70	Gain	8.224	0.004	0.75	0.83	0.82	0.77	0.79	0.58	4.50	0.30
OR2W1	0.743	2.63	Loss	12.000	0.001	0.96	0.65	0.74	0.94	0.81	0.62	2.78	0.06
OR5B17	0.750	2.81	Loss	6.750	0.027	0.58	0.92	0.88	0.69	0.75	0.50	7.00	0.45
OR11G2	0.771	2.97	Loss	6.171	0.013	0.67	0.83	0.80	0.71	0.75	0.50	4.00	0.40
OR6C65	0.781	2.53	Loss	6.171	0.013	0.67	0.83	0.80	0.71	0.75	0.50	4.00	0.40
OR11A1	0.785	3.09	Loss	6.000	0.014	0.75	0.75	0.75	0.75	0.75	0.50	3.00	0.33
ANN (all ORs)	1.000	0.53		24.000	<0.001	0.96	0.96	0.96	0.96	0.96	0.92	25.00	0.04

**Table 4 ijms-24-16681-t004:** Olfactory Receptors Hazard Ratio of Preeclampsia Development. Preeclampsia hazard ratio with 95% confidence interval (CI) of loss in OR expression before 20 weeks of gestational and the risk of early preeclampsia appearance. The low expression of *OR5H1*, *OR5T3*, *OR51A7*, *OR10J5*, and *OR6C6* showed an increased risk of developing PEC and the overexpression of *OR51B6* and *OR2AG2* was protective. The *p*-value was calculated with the Cox regression.

Olfactory Receptor	Hazard Ratio	95%CI	*p*-Value
*OR5H1*	6.08	1.32–27.96	0.021
*OR5T3*	7.42	2.12–26.01	0.002
*OR51A7*	6.17	1.34–28.39	0.019
*OR51B6*	0.18	0.06–0.59	0.004
*OR10J5*	3.89	1.22–12.43	0.022
*OR6C6*	7.99	1.03–62.21	0.047
*OR2AG2*	0.141	0.03–0.53	0.004

## Data Availability

Data available on request due to restrictions eg privacy or ethical. The data presented in this study are available on request from the corresponding author. The data are not publicly available due to the unfinished project involving these data.

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
