# Peer review of "Exploring the Potential of Olfactory Receptor Circulating RNA Measurement for Preeclampsia Prediction and Its Linkage to Mild Gestational Hypothyroidism"

_ijms, 2023, doi:10.3390/ijms242316681_

Round 1

Reviewer 1 Report

Comments and Suggestions for Authors

Dear authors!

I carefully studied your article. An article with an unusual approach to the diagnosis of preeclampsia and establishing a relationship with hypothyroidism. The olfactory is phylogenetically a very important in n the animal world. Of particular relevance of the problem of olfactory disturbancy was acquired in connection with Covid-19. Although  decrease in smell in humans is considered to be mainly more social than medical problem. This article demonstrates the importance of impairment in connection with various diseases, including a pathology of pregnancy.

Comments   References 30 and 31 should be done according to the rules of the journal. Graphs well reflect the results of the work.

P. 4 line 185Correction should be made.

The connection with hypoparathyroidism is less clear. But this probably does not diminish its importance.

An article with an unusual approach. It seems to me that we should make  more сlear the Сonclusion and the Discussion.

After eliminating corrections and making additions, the article may be published in the Journal

Comments on the Quality of English Language

The article is well English written.

Reviewer 2 Report

Comments and Suggestions for Authors

This work aimed to reveal the mechanism through which preeclampsia and gestational hypothyroidism occur by analyzing gene expression differences between healthy pregnancies and those with pregnancy complications. The investigators hypothesized that the activity of the olfactory receptor genes is underlying the often comorbidity of the two conditions. They identified DEG between conditions and controls, and between the two different conditions. They also reported overlapping DEGs among the OR superfamily that might count for the shared mechanism. In a separate cohort comprising 12 healthy pregnancies and 12 preeclampsia cases, the investigators identified individual OR that might be predictive of the onset of preeclampsia. Congratulations to the authors for completing a work attempting to address an important unanswered question.

There are three main weaknesses of the study as far as I am concerned. From section 3.4, it reads to me that the authors selected the ORs and evaluated their predictive power in the same 12 control + 12 preeclampsia blood samples. This resulted in overfitting and the ORs selected are unlikely to be useful in the general pregnancy population. Second, there was low statistical power in detecting DEG and in establishing the prediction of preeclampsia due to small sample sizes, diminishing how much the results can be trusted. Lastly, there is a weakness in the strength of the OR findings due to a lack of consideration of the large size of the OR family in relative to other gene sets (see detail below). 

The olfactory receptor superfamily is the largest in the human genome. This means that simply by chance, it is more likely to be identified as having some link with a disease or molecular process of interest regardless of whether it carries true biology or not. It is also empirically true that many findings in the genetics association field mapped to OR genes don’t replicate nor receive any support in functional work. The observed overlap between PEC and GHT DEGs among the OR genes could be also explained by the simple fact that there are many OR genes than other gene sets instead of OR is a shared feature between the two conditions. The investigators need to provide evidence against the former. One suggestion would be to take random gene sets of different sizes and examine if the size influences the proportion of overlapping. 

Please provide the rationale in the manuscript as to why OR6X1 and OR4E2 were selected among all ectopic ORs. 

How was the comparability of the GHT, PEC, and HTP ensured? Were they matched on any clinical or demographic factors? In the DEG analysis, were those factors included as covariates?

What is the chronological order of the onset of preeclampsia and gestational hypothyroidism? Does one usually precede the other?

Figure 1B - the heatmap is confusing in that it lacks the necessary description. What are across the rows? What does the color shade represent?

Table 1 is missing from the current manuscript. Please add.

Figure 2: boxplot can be confusing when the number of samples is small. Please consider using adding all dots, not just outliers. 
